# Invasive Fungal Diseases in Africa: A Critical Literature Review

**DOI:** 10.3390/jof8121236

**Published:** 2022-11-22

**Authors:** Felix Bongomin, Bassey E. Ekeng, Winnie Kibone, Lauryn Nsenga, Ronald Olum, Asa Itam-Eyo, Marius Paulin Ngouanom Kuate, Francis Pebalo Pebolo, Adeyinka A. Davies, Musa Manga, Bright Ocansey, Richard Kwizera, Joseph Baruch Baluku

**Affiliations:** 1Department of Medical Microbiology and Immunology, Faculty of Medicine, Gulu University, Gulu P.O. Box 166, Uganda; 2Department of Medical Microbiology and Parasitology, University of Calabar Teaching Hospital, Calabar P.O. Box 540281, Nigeria; 3Department of Medicine, School of Medicine, Makerere University, Kampala P.O. Box 7072, Uganda; 4Department of Medicine, School of Medicine, Kabale University, Kabale P.O. Box 317, Uganda; 5Department of Medicine, St. Francis’s Hospital Nsambya, Kampala P.O. Box 7176, Uganda; 6Department of Internal Medicine, University of Calabar Teaching Hospital, Calabar P.O. Box 540281, Nigeria; 7Department of Microbiology and Parasitology, Faculty of Science, University of Buea, Buea P.O. Box 63, Cameroon; 8Department of Reproductive Health, Faculty of Medicine, Gulu University, Gulu P.O. Box 166, Uganda; 9Department of Medical Microbiology and Parasitology, Olabisi Onabanjo University Teaching Hospital, Sagamu P.O. Box 121102, Nigeria; 10Department of Environmental Sciences and Engineering, Gillings School of Global Public Health, University of North Carolina at Chapel Hill, 4114 McGavran-Greenberg, 135 Dauer Drive, Chapel Hill, NC 27599, USA; 11Division of Evolution, Infection and Genomics, Faculty of Biology, Medicine and Health, University of Manchester, Manchester M13 9PL, UK; 12Translational Research Laboratory, Department of Research, Infectious Diseases Institute, College of Health Sciences, Makerere University, Kampala P.O. Box 22418, Uganda; 13Division of Pulmonology, Kiruddu National Referral Hospital, Kampala P.O. Box 7178, Uganda; 14Makerere Lung Institute, College of Health Sciences, Makerere University, Kampala P.O. Box 22418, Uganda

**Keywords:** invasive fungal diseases, histoplasmosis, cryptococcosis, histoplasmosis, pneumocystosis, emergomycosis, invasive candidiasis, invasive aspergillosis, mucormycoses, Africa

## Abstract

Invasive fungal diseases (IFDs) are of huge concern in resource-limited settings, particularly in Africa, due to the unavailability of diagnostic armamentarium for IFDs, thus making definitive diagnosis challenging. IFDs have non-specific systemic manifestations overlapping with more frequent illnesses, such as tuberculosis, HIV, and HIV-related opportunistic infections and malignancies. Consequently, IFDs are often undiagnosed or misdiagnosed. We critically reviewed the available literature on IFDs in Africa to provide a better understanding of their epidemiology, disease burden to guide future research and interventions. Cryptococcosis is the most encountered IFD in Africa, accounting for most of the HIV-related deaths in sub-Saharan Africa. Invasive aspergillosis, though somewhat underdiagnosed and/or misdiagnosed as tuberculosis, is increasingly being reported with a similar predilection towards people living with HIV. More cases of histoplasmosis are also being reported with recent epidemiological studies, particularly from Western Africa, showing high prevalence rates amongst presumptive tuberculosis patients and patients living with HIV. The burden of pneumocystis pneumonia has reduced significantly probably due to increased uptake of anti-retroviral therapy among people living with HIV both in Africa, and globally. Mucormycosis, talaromycosis, emergomycosis, blastomycosis, and coccidiomycosis have also been reported but with very few studies from the literature. The emergence of resistance to most of the available antifungal drugs in Africa is yet of huge concern as reported in other regions. IFDs in Africa is much more common than it appears and contributes significantly to morbidity and mortality. Huge investment is needed to drive awareness and fungi related research especially in diagnostics and antifungal therapy.

## 1. Introduction

Invasive fungal diseases (IFDs) are a major cause of morbidity and mortality, especially in immunocompromised individuals, such as those with human immunodeficiency virus (HIV), haematological malignancies, organ transplant recipients, as well as prolonged immunosuppressive therapy [1,2,3]. The majority of IFDs occur as opportunistic infections and are defined as the presence of fungal elements in deep tissues of biopsy or needle aspirates identified on culture or histopathological investigations [4]. IFDs still trammel developing countries, with high burden of HIV, tuberculosis, and poverty being the three main drivers [5].

In addition, timely diagnosis of IFDs is a challenge in Africa due to unavailability of reliable point-of-care tests (POCTs), with barriers, such as high cost of tests, lack of awareness among healthcare providers, delays, and low sensitivity of confirmatory blood cultures [5,6]. Early diagnosis and initiation of appropriate antifungal therapy is important in the eradication of IFDs with eventual reduction in morbidity and mortality due to IFDs [7,8].

However, there is a paucity of data on the burden of invasive fungal diseases in Africa. Therefore, the overarching aim of this critical review is to highlight the current state of the burden of invasive fungal diseases in Africa through comprehensive literature review of published cases of IFDs, including cryptococcosis, histoplasmosis, aspergillosis, pneumocystis pneumonia, candidaemia, mucormycosis, talaromycosis, emergomycosis, blastomycosis, coccidiodomycosis, paracoccidiodomycosis, chromoblastomycosis, and sporotrichosis, in the continent.

## 2. Methods

We conducted a literature search using PubMed, Google Scholar, and African Journal Online to identify published papers on invasive fungal infections from Africa. No date limitation or any other search criteria were applied, to avoid the exclusion of articles on IFDs in Africa. The following search terms were used: ‘histoplasmosis and Africa’, ‘cryptococcosis and Africa’, ‘aspergillosis and Africa’, ‘blastomycosis and Africa’, ‘pneumocystis pneumonia and Africa’, ‘candidiasis and Africa’, ‘mucormycosis and Africa’, ‘emergomycosis and Africa’, ‘talaromycosis and Africa’, ‘blastomycosis and Africa’, ‘sporothricosis and Africa’, ‘coccidiodomycosis and Africa’, and ‘paracoccidiodomycosis and Africa’. All authors were involved in initial data curation, thereafter 3 authors (FB, BEE, WK) screened publications for eligibility. We included retrospective studies, prospective studies, and case series predominantly. Case reports were included for IFDs with scant reports. References in all relevant papers were also reviewed for additional publications (‘snow balling’) on IFDs that may not have been published in the searched databases. Publications without patients’ country of origin were excluded. Publications on IFDs outside Africa were excluded. Data extracted from each case included: age, gender, disease type (single focus vs. multiple foci), sites of infection, clinical features, diagnostic test, treatment, and patient outcome.

## 3. Results

### 3.1. Cryptococcosis

Cryptococcosis is an opportunistic fungal infection caused by the fungal species *Cryptococcus neoformans* and *Cryptococcus gattii* [9,10]. It is the leading cause of morbidity and mortality in AIDS patients worldwide, affecting one million people annually [9,10]. The invasive form of the disease, cryptococcal meningitis (CM) kills more than 180,000 HIV-positive patients annually, with over 70% in low-income countries, including sub-Saharan African (SSA) countries [11,12]. Rare forms of extra-neuromeningococcal cryptococcosis exist in Africa [13]. The rate of CM is higher than tuberculosis meningitis [14]. Typical signs and symptoms associated with the disease are headaches, neck stiffness, and altered consciousness [15,16,17,18]. Low body mass index, low haemoglobin, and low CD4 cell count (<200 cells/mm^3^) are significant predictors for CM [19,20,21,22,23,24,25]. The burden of cryptococcosis has previously been estimated across Africa, and it remains extremely high [26]. It was estimated at 720,000 (144,000–1.3 million) and 162500 (113,000–193,000) cases annually in 2009 and 2014, respectively, in patients with CD4 count < 100 cells/mm^3^ in SSA [27,28], Table 1. A review carried out in 2020 based on cross-sectional studies estimated the magnitude of cryptococcosis in SSA countries at 8.3% (6.1–10.5%) [29].

A recent global burden of cryptococcosis estimated CrAg positivity and cryptococcal meningitis at 75,000 (55,000–95,000) and 63,000 (45,000–80,000) cases, respectively, in Eastern and Southern Africa, and at 22,000 (19,000–26,000) and 19,000 (16,000–22,000) cases, respectively, in Western and Central Africa. The burden of CM in some African countries has been estimated over the years, with the lowest rate in Algeria (35 cases/year) and the highest in Nigeria (57,866 cases/year) [30]. These estimates are based on studies carried out in these countries and usually in HIV-positive patients with low CD4 count [6,31,32,33,34,35,36,37,38,39,40,41,42,43,44,45,46] (Table 1). The availability of studies and data on cryptococcal antigenemia and CM varies from one country to the other (Table 2). A multicenter study estimated the prevalence of cryptococcal antigenemia in four regions of Nigeria at 2.3% (1.8–3%) in HIV patients with CD4 < 200 cells/mm3 [47]. However, this prevalence significantly differs from one region to another [47], with a prevalence from 1.4% (4/290) to 19.67% (59/300) for cryptococcal antigenemia [9,14,23,24,25,48,49,50,51], and from 16.8% (31/184) to 36% (36/100) for CM [10,52]. In Uganda, a prospective study found 32 (5.7%) HIV-infected patients with cryptococcal antigen (CrAg) positivity from 2009 to 2010 [53]. Still in Uganda, in two cross-sectional studies in HIV-positive patients, prevalence of 5.7% and 19% were obtained [22,53]. Jacinta et al. (2012) also found 6.5% cases of CM [22]. A large screening program in South Africa from 2017 to 2019 among HIV patients with low CD4 count found a cryptococcal antigenemia rate of 5.8% (35,000/600,000) [54]. A prevalence of 23.1% (43/186) for cryptococcal antigenemia and 21.7% (5/23) for CM were obtained from a similar study in Cameroon [55]. Little is known about the prevalence of cryptococcosis in Ethiopia. This prevalence ranges from 4% to 11.43% with CD4 < 100 cells/mm3 as a common factor [56,57,58,59] and sometimes co-infection with other diseases [56,57]. More data on cryptococcosis are available in Mali, Democratic Republic of Congo (DRC), Togo, Kenya, Ghana, Nigeria, Burkina Faso, Botswana, Senegal, Tanzania, South Africa, Cameroon, and Mozambique, sometimes with a high mortality rate [13,16,17,18,60,61,62,63,64,65,66,67,68,69,70,71,72,73,74,75]. Data on cryptococcosis in HIV negative patients and children is rare. Then, rates of 18.8% (22/117), 2.5% (1/150), 7% (16/228), and 1.47% (3/204) for CM and cryptococcal antigenemia in HIV-negative patients were obtained from studies in Nigeria and Mali [10,49,51,76]. Two studies in Cameroon determined a prevalence in children of 6.12% (9/147) and 3.6% (12/331) for cryptococcal antigenemia and CM, respectively [67,69]. Data in some countries are available only in form of case reports [77], Table 2.

**Table 1 jof-08-01236-t001:** Estimated burden of cryptococcosis in some African countries.

Country	Pub Year	Burden	Rate/100K	Prevalence Used for Estimation	Group at Risk	References
Senegal	2015	366	NA	7%	HIV/AIDS	Badiane et al. [31]
Burkina Faso	2018	459	2.5	3.4%	HIV/AIDS	Bamba et al. [39]
Ethiopia	2019	9900	9.4	11.7%	HIV/AIDS	Tufa and Denning [40]
Togo	2021	1342	18.52	6.12%	HIV/AIDS	Dorkenoo et al. [41]
Namibia	2019	543	21.8	3.3%	HIV/AIDS	Dunaiski and Denning [42]
Mozambique	2018	18,640	70.5	19.4%	AIDS	Sacarlal and Denning [43]
Ghana	2019	6275	21.7	12.7%	HIV/AIDS	Ocansey et al. [45]
Morocco	2022	160	0.43	2.9%	AIDS	Lmimouni et al. [46]
Côte d’Ivoire	2020	4590	18.22	12.7	HIV/AIDS	Koffi et al. [36]
Algeria	2016	36	0.09	5.6%	HIV/AIDS, Cancer	Talbi and Denning [6]
Egypt	2017	38	0.0	NA	HIV/AIDS	Zaki and Denning [32]
Cameroon	2018	6720	30	11%	HIV/AIDS	Mandengue and Denning [33]
Nigeria	2014	57866	37.4	10% of new adults AIDS cases 12.7% of adults with CD4 < 200 10% cases in children	HIV/AIDS	Oladele and Denning [36]
Uganda	2013	2783	NA	5.8%	HIV/AIDS	Parkes-Ratanshi and Denning [34]
South Africa	2019	8357	14.8	NA	HIV/AIDS	Schwartz and Denning [37]
Kenya	2016	11,900	29	7%	HIV/AIDS	Guto et al. [44]
Zimbabwe	2021	6086	41	NA	HIV/AIDS	Pfavayi et al. [35]

NA: not available.

### 3.2. Histoplasmosis

Histoplasmosis is a serious fungal disease endemic in the Ohio and Mississippi river valleys in the United States, as well as Central, South America, Western Africa, Southern Africa, Eastern Africa, Central Africa, and Southeast Asia [79,80,81,82]. The classical form of the disease is caused by *Histoplasma capsulatum* var. *capsulatum* (Hcc), while the African type is caused by *Histoplasma capsulatum* var. *duboisii* (Hcd) [79,80]. *Histoplasma* infection is commonly acquired via the inhalation of microconidia. Its greatest attributable risk factor in the adult population is HIV/AIDS and was classified as an AIDS-defining illness in 1987 [79]. However, retrospectively, several cases of histoplasmosis were also recorded in Africa prior to HIV/AIDS pandemic [79]. On the contrary, in the paediatric population, histoplasmosis is predominantly associated with risk factors other than HIV, including environmental exposures and toxins, autoimmune diseases, childhood malignancies, as well as their treatment, lung diseases, immunosuppressive therapies, pancytopenia, T-cell deficiency, and malnutrition [83,84]. Clinical features are non-specific and mimic other clinical entities, including tuberculosis, malignancies, tropical splenomegaly syndrome, leishmaniasis, amongst others [85,86,87]. The classical form usually presents as a pulmonary disease, while the African-type presents with extrapulmonary manifestations, including bone lesions and ulcers [79,80]. Diagnosis of histoplasmosis often requires a high index of clinical suspicion otherwise may lead to delayed or misdiagnosis. The gold standard for diagnosis is culture, however most cases are diagnosed by histopathology, as fungal cultures are not routinely available in many African countries. Other diagnostic modalities include *Histoplasma* antigen assay, antibody detection, molecular techniques, direct examination, and the use of peripheral blood smear [79,80].

Despite the apparent increase in reported cases, the true burden of histoplasmosis in Africa is yet unclear due to several reasons: (1) poor awareness on the part of clinicians with some cases diagnosed post mortem [85,86,87]; (2) a low index of suspicion on the part of clinicians resulting in several cases being misdiagnosed as other clinical entities [79,85,86,87]; (3) inadequate or the lack of diagnostic capacity across African countries [88,89]; (4) data on the incidence and prevalence, as well as information on its morbidity and mortality in most African countries are fragmentary and perhaps not available in some African countries [79]. Be that as it may, some studies have described cases in some specific countries in Africa and across Africa generally. One review by Oladele et al. that spanned six decades (1952–2017) identified 470 cases of histoplasmosis with HIV-infected patients consisting of 38% (178) of the cases. West Africa had the highest number of recorded cases with 179; the majority (*n = * 162 cases) were caused by *Histoplasma* (*H*) *capsulatum* var. *dubuosii* (Hcd). In total, 150 cases were reported from the Southern African region, and the majority (*n* = 119) were caused by *H. capsulatum* var. *capsulatum* (Hcc). Hcc was found to be the most predominant infective agent in Africa, while Hcd was primarily found in Central and West Africa and Madagascar. Most of the reported cases from Africa were diagnosed by culture and histology; only in five countries (Tanzania, Benin, South Africa, Egypt, and Uganda) was serology reported as being used to make a diagnosis, and in three of the cases, the samples were processed in Western countries [79]. A more recent review focused on African histoplasmosis in the context of HIV/AIDS by Develoux et al. identified 94 well documented cases (1993 to 2019) of Hcd infection, with 30.1% of the patients under 18 years old. HIV coinfection rate was 20.8% with fever, lymphadenopathies, and absence of bone infection being the differentiating elements from patients living without HIV [90]. With regard to the paediatric population in Africa, a global review (1939–2021) mentioned 65 cases with most of the cases reported from the Republic of Congo (*n* = 26) and Nigeria (*n* = 13) [83]. Another review (1950–2021) yet focused on histoplasmosis in the paediatric population in Africa described 44 selected cases distributed across Western Africa (38.6%, *n* = 17), Eastern Africa (9.1%, *n* = 4), Southern Africa (9.1%, *n* = 4), and Central Africa (43.2%, *n* = 19). No case report was found from Northern Africa. The age range was 1–17 years, with a mean of 9.2. Of the 44 case reports, 8 cases (18.2%, 8/44) were caused by *Histoplasma capsulatum* var. *capsulatum*, 33 cases (75%, 33/44) were caused by Histoplasma *capsulatum* var. *duboisii*, and specie identification was not found in 3 cases. Only three (6.8%) cases were HIV positive; 56.8% (25/44) were disseminated histoplasmosis, pulmonary histoplasmosis accounted for just one case (2.3%, 1/44) [86]. Findings from more recent studies and other reviews focused on specific countries in Africa are as summarized in Table 3 [83]. None of the studies indicated case fatality rates.

### 3.3. Invasive Aspergillosis

Invasive aspergillosis (IA) is a life-threatening infection that affects those with impaired immune systems [99]. It is caused by *Aspergillus* species, which are saprophytic fungi that belong to the class ascomycetes and produce conidia that are easily dispersed into the air [100]. There are more than 300 species of *Aspergillus*, and 90% of human infection is caused by *A. fumigatus* with the other implicated species being *A. flavus*, *A. terreus*, *A. niger*, and *A. nidulans* [101]. The lungs are the common site of infection and the conidia when inhaled results in a diverse clinical spectrum namely, allergic, chronic, and invasive aspergillosis [101].

Globally, IA affects more than 300,000 people annually and the fatality rate ranges from 30 to 80% [102]. The incidence rises with an increase in immunosuppression [102] because the phagocytic and neutrophilic functions of the immune system that prevent conidia germination are impaired [103]. Hence, IA is commonly found in those with neutropenia, recipient of transplantation, steroid therapy, and chronic granulomatous disease. Other people that could be affected are those who recently underwent surgery; are in intensive care units; those with HIV/AIDS, coronavirus disease 2019 (COVID-19), influenza virus, diabetic mellitus, chronic obstructive pulmonary disease, and cytomegalovirus; and those on newer immunosuppressive agents, such as tumour necrosis factor-alpha inhibitors [104]. The most common form is invasive pulmonary aspergillosis (IPA). Diagnosing IA is challenging because it involves culture of the respiratory specimen, histology of biopsy tissue, non-culture-based methods using *Aspergillus* antigen, imaging with computer tomography, magnetic resonance imaging, and molecular techniques with polymerase chain reaction [105].

In Africa, the burden of IA was estimated as 0.05 to 10.9 per 100,000 population. However, most of these estimates do not take into consideration the “at-risk population” and the clinical spectrum of invasive aspergillosis. The estimates per 100,000 population were 4.8 to 16 in Southern Africa, 0.05 to 10.9 in Northern Africa, and 0.3 to 6.25 in Western Africa while it ranges from 7.1 to 10.7 and 3.2 to 6.9 in Northern Africa and Central Africa, respectively [106].

The few cases published in Africa are listed in Table 4 below. These include six case reports, and four prospective and one retrospective study. Of the case reports, the underlying risks were HIV, recipients of renal transplant, antibiotic use, cytomegalovirus, diabetics mellitus, renal failure, and chronic granulomatous disease, while none was specified from the cases reported from Sudan. From Nigeria, a case in an immunocompetent elderly woman in whom the diagnosis was made with computer tomography (CT) was reported. In all the cases reported below, galactomannan assay in combination with CT and/or culture was only performed in the cases reported from Tunisia. Concerning the observational studies, IPA was the only clinical spectrum reported and the underlying risk were Hodgkin lymphoma; HIV; haematological malignancies, such as acute myeloid leukemia (AML), acute lymphoid leukemia (ALL), and neutropenia; broad-spectrum antibiotics; severe combined immunodeficiency; and bronchiectasis. In three of the cases, the diagnosis was established by histopathology of biopsy specimen after post-mortem.

The data on invasive aspergillosis are limited in Africa, and this could have been because of the low level of clinicians’ awareness of invasive fungal infections (IFI) and the few numbers of medical mycologist specialists. Importantly, diagnostic tools, such as *Aspergillus* antigen assays are not readily available; also, the high prices involved in carrying out CT scans and polymerase chain reaction (PCR) are contributory factors that hamper diagnosis, leading to the underreporting of cases.

### 3.4. Pneumocystis Pneumonia

*Pneumocystis jirovecii* is a fungus belonging to the phylum Ascomycota and is famed for causing a fatal infection of pneumonia among immunocompromised individuals [117]. Initially classified as a protozoan, rRNA sequencing led its subsequent designation to the fungi kingdom [117]. The HIV epidemic brought *P. jirovecii* to the forefront as a cause of a severe form of pneumocystis pneumonia (PCP) among people with advanced HIV. As such, most epidemiological data available from Africa are among people living with HIV (PLWH). About 15% of new AIDS patients develop PCP.

The burden of PCP among PLWH in Africa is well described in two systematic reviews. In a meta-analysis of hospital-based studies from 18 countries in sub-Saharan Africa, the overall prevalence of PCP was 15.4% but it was higher among people with respiratory symptoms (18.8%), inpatients (22.4%), and inpatients with respiratory symptoms (24%) [118]. This meta-analysis demonstrated a decline in the prevalence of PCP among inpatients over the years: 28% in the 1990s, 27% between 2000 and 2004, and 9% after 2005 [118]. This trend correlated with the proportion of people initiated on anti-retroviral therapy (ART) and cotrimoxazole prophylaxis. The rate of respiratory co-infections (alongside *P. jirovecii*) was reportedly high (29.3%) and included pulmonary tuberculosis (14.8%), bacterial pneumonia (18.7%), pulmonary cytomegaly virus (3.9%), and pulmonary cryptococcosis (1.4%) [118]. In most African countries, PCP is diagnosed clinically and managed with high dose cotrimoxazole without laboratory confirmation. Radiology is non-specific. PCR is considered the gold standard for diagnosis. However, the low diagnostic accuracy of clinical and radiological features for PCP limits an accurate estimation of the true burden of PCP in Africa where laboratory confirmation by microscopy and polymerase chain reaction (PCR) on bronchoalveolar lavage is largely inaccessible in many places. A more recent meta-analysis of studies from 15 African countries reported an overall prevalence of laboratory-confirmed *P. jirovecii* of 19% on any respiratory sample from adult PLWH with respiratory symptoms [119]. The prevalence varied from 15% among studies that utilised microscopy to 22% among studies that used PCR. Interestingly, the prevalence of laboratory-confirmed *P. jirovecii* has remained relatively level in the pre-ART era (1995–2005) and the ART era (2006–2020) at 21% and 18%, respectively [119]. The case fatality rate from PCP among PLWH has been estimated to be 18.8% [118].

With regard to *P. jirovecii* colonisation (a positive *P. jirovecii* PCR test in the absence of respiratory symptoms), three small studies from Tanzania, Guinea-Bissau, and Cameroon reported a prevalence of 0.3%, 3%, and 42.9% (18.9% among HIV-negative individuals), respectively, among PLWH [120,121,122]. Although the clinical relevance of *P. jirovecii* colonisation is not very apparent, colonisation could increase transmission of *P. jirovecii* and disease progression among at-risk individuals and has been implicated in chronic obstructive pulmonary disease exacerbations [123].

There is a paucity of data on the burden of *P. jirovecii* among HIV-uninfected individuals in Africa. HIV-negative people with haematological malignancies, long-term steroid use, people on treatment for solid cancers with chemo-radiotherapy, recipients of organ transplants and people with rheumatological conditions are at risk of PCP [124]. However, systematic reviews on *P. jirovecii* in non-HIV populations report no studies from Africa [125,126]. Nonetheless, a few studies among children exist. The prevalence of *P. jirovecii* was 12.8% in one study among HIV-negative children admitted with hypoxic pneumonia in South Africa [127]. In another multi-centre study, the proportion of severe pneumonia caused by *P. jirovecii* requiring hospitalisation among children was reported to be 0.2% in The Gambia, 2.4% in Mali, 2.3% in Kenya, 4.1% in Zambia, and 2.1% in South Africa [128]. More studies are needed to characterise disease caused by *P. jiroveciii* among HIV-uninfected adults in Africa.

### 3.5. Candidemia

Candidaemia refers to the presence of *Candida* species in the blood stream and is among the most common blood-stream infections [129]. The most common *Candida* species that cause candidaemia include: *C. albicans*, *C. auris*, *C. krusei*, *C. parapsilosis*, *C. tropicalis,* and *C. glabrata* [130]. Diagnosis of candidaemia is by blood culture. A positive 1,3-beta –D-glucan (BDG) assay is suggestive but is, however, not specific, as it has been shown to be positive for other fungal diseases, including cryptococcosis and pneumocystis pneumonia. A study on the comparative sensitivity of 1,3 beta-D-glucan for common causes of candidaemia showed an overall sensitivity of 79% and per species sensitivity of 81%, 72%, 90%, 71%, and 100% for *C. albicans, C. parapsilosis, C. glabrata, C. auris,* and *C. krusei*, respectively [131]. In a recent study, Lockhart et al. showed that *C. auris* consisting of clades (I to IV) is an emerging healthcare associated pathogen with high mortality due to limited treatment options as a result of resistance to antifungal drugs, as well as hygiene and infection control measures, such as hospital chlorine-based disinfectants [132,133]. In fact, the high nosocomial transmission of *C. auris* has been attributed to the antifungal and disinfectant resistance, use of reusable skin surface temperature monitoring probes, and prior exposure to systemic fluconazole [133,134]. A recent publication by Parak et al. revealed 45 cases of *C. auris* caused candidaemia in South Africa, all of which were susceptible to amphotericin B and micafungin however, patients treated with amphotericin B only had a higher mortality rate than those treated with an echinocandin [135].

*C. albicans* has been shown to have higher mortality than *C. auris* (36% and 29%, respectively) [136]. A rising number of cases of antifungal resistance by *Candida* species has been registered with the most recent retrospective study conducted in South Africa showing an overall rise in the proportion of azole resistant candidaemia cases from 39.6% in 2016 to 69.5% in 2020 [137]. However*,* only 0.2% and 0.3% of *C. albicans* and, 0.9% and 6.6% of non-albicans isolates were resistant to amphotericin B and echinocandin, respectively [137].

Recent publication from South Africa showed through laboratory based surveillance that neonates were the most affected age group (49%) followed by infants (27%) with *C. parapsilosis* (42%) and *C. albicans* (36%) as the commonest causative agents of single species candidaemia [138]. Moreover, the highest 30-day in-hospital mortality was identified to be highest in neonates (43%) and adolescents (43%) [138]. Hegazi et al. in a study conducted in Egypt showed a slightly lower mortality (16.7%) due to candidaemia among patients aged 6 months to 15 years [139]. A five-year retrospective descriptive study conducted in South Africa by Hussain et al., showed a high overall mortality associated with candidaemia with 55% of the participants dying from candidaemia and its complications [129].

Surveillance studies in South Africa, Algeria, and Kenya, reports *C. parapsilosis*, *C. tropicalis,* and *C. auris* as the most commonly isolated species, respectively [136,138,140]. Other surveillance-based studies conducted in 2009 and 2010 in South Africa, and retrospective hospital-based studies in South Africa and Nigeria, found *C. albicans* to be the most cause of candidaemia (23.8%, 73%, and 77%, respectively) [134,141,142]. As evidenced by results from a prospective case–control study in Cairo, Egypt, neutropenia is highly associated with a diagnosis of candidaemia for which, 56% and 16% of premature neonates with neutropenia and without neutropenia, respectively, were diagnosed with candidaemia based on blood culture results [143]. Table 5 summarizes published cases of candidaemia in Africa.

### 3.6. Mucormycosis

Mucormycosis is an opportunistic invasive disease predominantly found in immunocompromised individuals [148]. The sequelae is severe and often life threatening [149,150].

These fungi are members of the genera *Mucor*, *Rhizopus*, *Rhizomucor*, *Syncephalostrum,* and *Lichtheimia* in the order *Mucorales* [149]. They are widely distributed in the environment as airborne spores which may result in skin and respiratory infections in susceptible individuals [149]. They are also seen as bread mold, laboratory contaminant, and is a common inhabitant of soil and decaying vegetation [151].

Immunocompromised states which result in invasive fungal diseases are: diabetes mellitus, recipients of haematopoietic stem cells or solid organ transplants, patients infected with HIV, and those receiving chemotherapy for cancer [148]. Penetrating trauma, steroid use, deferoxamine therapy, burns, and complications as a result of healthcare procedures are also implicated [152].

There are varied sites of involvement of mucormycosis infection with rhino–sino–orbital–cerebral, pulmonary, cutaneous, and gastrointestinal involvement being the most common sites of involvement reported [148,151,153]. There is a great need for a high index of suspicion to aid in early identification and treatment of this disease condition as it is associated with high mortality rate with survival rate as low as 3% reported in the absence of treatment [154]. Prompt commencement of treatment is required to reduce the mortality rate even though in the presence of appropriate medical management, mortality is still high [152]. Treatment involves a combination of surgery to completely remove the infected tissues and first line antifungal therapy, which are amphotericin B, Isavuconazole, and Posaconazole [155]. In addition to this, identification and treatment of a predisposing factor is essential in order to further reduce mortality rate [156].

In Africa, Mucormycosis infection has been documented to span across all age groups. A case series performed in Egypt by El-Mahallawy et al. [157], documented mucormycosis in the paediatric age group with patients age ranging from 1.5 years to 12 years on treatment for haematological malignancies: acute lymphoblastic leukaemia and acute myeloid leukaemia, and a high mortality recorded at 60% [157]. Mucormycosis has also been recorded in a 10-month-old baby in South Africa attributed to underdeveloped immunity [151]. Its presentation as recorded in case series of rhinocerebral mucormycosis by Bodenstein et al. [158], Hauman et al. in South Africa [151]; Zaki et al. [152], Alfishawy et al. [159], Alloush et al. in Egypt [160]; and Anane et al. [161] in Tunisia. Its presentation as gastrointestinal mucormycosis was recorded by Thomson et al. [162] and Kahn et al. [163] in South Africa. Its presentation as pulmonary mucormycosis was recorded by Feki et al. [164] in Tunisia; and Madeney et al. [165] and El Mahallawy et al. [157] in Egypt. In addition, there are also case reports in this regard [166,167,168,169].

There is a notable increase in the occurrence of murcormycosis in Africa which corresponds to the increase global prevalence of diabetes mellitus and immunocompromised patients [148,153] and also the emergence of the COVID-19 pandemic. Case series from Egypt, performed by Abd El-hameed et al. [166], Alfishaway et al. [159], and Alloush et al. [160] revealed the bidirectional relationship between COVID-19 and diabetes mellitus with increased risk of mucormycosis, while new onset diabetes (NOD) and the exacerbation of pre-existing diabetes mellitus had been identified in COVID-19 patients [159,166].

In Tunisia, Trabelsi et al. reported 11 cases (3.4%) out of 321 renal transplant recipients with a high mortality rate of 72% [108]. This reiterates the need for heightened suspicion of this fungal disease entity in renal transplant patients as Africa currently has 46 countries involved in renal transplant [168]. A 70-year retrospective study conducted in Uganda also revealed four cases of mucormycosis due to unidentified fungi that were diagnosed by PAS stain and microscopy and affecting multiple body parts [98]. Diagnoses is mostly by histopathology [151,152,157,158,162,163,164,165]. Other modalities include microscopy [151,159,160,161], culture [152,157,159,164,165], and PCR [152], Table 6.

Problems faced across Africa in the diagnosis and management of mucormycosis infection include: late presentation of patients [151], its presumed rarity leading to delay in diagnosis [151], and the poor accessibility to antifungal medications for management of patients [169].

Based on widespread areas of involvement of invasive mucormycosis and associated increased mortality, a greater surveillance protocol set up is needed in African countries to prevent and reduce morbidity and mortality in affected individuals.

### 3.7. Talaromycosis

Talaromycosis, previously known as penicilliosis, is a fungal infection caused by the fungus *Talaromyces marneffei* (*Penicillium marneffei*). It is endemic in east and southeast Asia and considered a neglected tropical disease [170]. It may present with painless skin lesions on the face and neck, fever, anaemia, large lymph glands, and liver [171]. It typically occurs in immunocompromised patients, such as those with HIV, cancer, organ transplant, long-term steroid use, old age, malnutrition, or autoimmune disease. *T. marneffei* is the only thermally dimorphic fungus in the genus *Talaromyces* [172]. As most dimorphic fungi are, it exists as a mold in the environment but forms small round yeast cells in host tissue. Data are limited about its natural habitat. However, it has been isolated from soil [171]. There is evidence showing that heavy rainfall may provide favourable conditions for the growth and dissemination of the fungus [173]. Infection is thought to follow inhalation of fungal spores from unidentified environmental sources. Incubation period may vary, and the fungus can sometimes cause an asymptomatic dormant infection for long periods [174]. Diagnosis is usually made by the identification of the fungus from clinical specimens, either by microscopy or culture. Biopsies of skin lesions, lymph nodes, and bone marrow demonstrate the presence of organisms on histopathology.

As seen in the table below, data are scant about the burden of talaromycosis in Africa. We found only two case reports [175,176], Table 7. From these reports, HIV and a travel history to Asia are the major risk factors. Cases are aged between 37 and 83 years. Most common clinical presentation includes multiple umbilicated papules associated with cough, fever, loss of appetite, loss of weight, urethral discharge, febrile pneumonia, dyspnoea, “molluscum contagiosum” such as lesions located on the face, arms, neck, and trunk. It is mainly diagnosed using culture-based methods in Africa. However, some sites employ molecular diagnostics, although these are rarely available in Africa. One case was managed with Itraconazole and responded well, while the other died within 12 h of admission.

### 3.8. Emergomycosis

Emergomycosis is a systemic fungal infection caused by the fungus *Emergomyces* (formerly called *Emmonsia*). *Emergomyces* is a dimorphic fungus and consists of five known species that have been reported globally [177]. These species are *Es. Pasteurianus*, *Es. Africanus*, *Es. Canadensis*, *Es. Orientalis*, and *Es. Europaeus*. *Es. pasteurianus* and *Es. africanus* are commonly isolated from Africa with South Africa having the highest number of cases [177]. *Es. africanus* is a newly discovered fungus within this group and, so far, has only been found in southern Africa. *Emergomyces* is also a thermally dimorphic fungus and is known to cause disease globally mostly in people with advanced HIV disease [178]. It is found in soil and human infection is through inhalation of fungal spores. So far, emergomycosis has been report on four continents: Asia, Europe, Africa, and North America. However, considering the increasing global burden of HIV, it is presumed that the disease must have a worldwide distribution with many cases going undetected [179]. Diagnosis is challenging and is confirmed by culture and/or histology. It should be considered in the differential diagnosis of histoplasmosis as there is considerable clinical and histopathological similarities between the two diseases. Currently, there are no consensus guidelines for the treatment of emergomycosis. Treatment requires antifungal medicines, such as amphotericin B for 1–2 weeks, followed by oral itraconazole for at least 12 months.

Data about Emergomycosis in Africa are limited and mostly reported in case reports/series, Table 8. The table below summarizes some of the published cases of Emergomycosis from Africa. Majority of the case were reported from South Africa. However, considering the burden of HIV in Africa, we believe that there are more cases across all Africa that go undetected due to low index of clinical suspicion and the lack of diagnostics. From these identified cases, the median age of patients is about 34 years. HIV is the major risk factor reported in all cases with majority having CD4 count less than 200 cells/mm^3^ and undetectable viral load. Most of the cases have skin lesions as the primary symptom. Other presentations include anaemia, pneumonia, gastroenteritis, herpes gingivostomatitis, and weight loss. Diagnosis was challenging and mainly performed using a combination of histology, culture, and molecular tests (mostly sequencing). These cases were mostly managed using amphotericin B and itraconazole. Some of the cases we identified in this article were managed with Fluconazole. However, in vitro susceptibility data from South Africa recommends the use of amphotericin B, followed by itraconazole, voriconazole, or posaconazole. Fluconazole was a relatively less potent agent [180].

Schwartz et al. summarized the geographic distribution, clinical characteristics, and management of 54 cases of disseminated emmonsiosis published across South Africa from January 2008 through to February 2015 [181]. Two more cases were described in South Africa by Heys et al. in 2014; one being immunocompetent and the other a renal-transplant patient receiving immunosuppressive therapy [182]. According to the authors, this one case of disseminated emmonsiosis in an apparently immunocompetent person raised the question of whether this was a more virulent strain or different species or whether the patient had exposure to a large amount inoculum or had an undiagnosed immune disorder.

*Es. africanus* has also been isolated from air samples from 34 days distributed over 11 weeks in Cape Town, South Africa [183]. It has also been isolated by PCR in 30% of soil samples from a wide range of habitats in South Africa [184].

**Table 7 jof-08-01236-t007:** Published cases of Talaromycosis in Africa.

Authors	Year	Country	Study Type	Number of Cases	Sex	Age	Risk Factors	Symptoms	Diagnostic Tool	Treatment	Outcomes
Guiguemde et al., [175]	2019	Burkina Faso	Case report	1	M	83	HIV, CD4 = 240 cells/Ul Poor ART adherence	Persistent itching skin lesions on the right foot, >1 year	Culture	Itraconazole (400 mg/day) for 8 weeks	Favourable
Govender et al., [176]	2014	South Africa	Case report	1	F	37	HIV, Cd4 = 20 cells/Ul travel to China	Skin lesions	Blood smear, β-D-glucan assay, Culture, Gene sequencing	NS	The patient died within 12 h of admission.

M, Male; F, Female; NS, Not stated.

**Table 8 jof-08-01236-t008:** Studies published on emergomycosis in Africa.

Authors	Year	Country	Study Type	Number of Cases	Sex	Age	Risk Factors	Clinical presentation	Diagnostic Tool	Causative Agent	Treatment	Outcomes
Kenyon et al., [185]	2013	South Africa	Case series	13	M = 8 F = 5	Median age = 34 years	HIV, median CD4 count = 16 cells/Ul	Anemia Skin lesions	DNA sequencing	*Emmonsia* species	amphotericin B Itraconazole	Death (n = 3), LTFU (n = 1), Favourable (n = 9)
Moodley et al., [186]	2019	South Africa	Case report	1	F	31	HIV, CD4 count = 80 cells/Ul	Skin lesions	Histopathology, Culture, Molecular testing	*Emergomyces africanus*	Fluconazole	Favourable
Schwartz et al., [187]	2017	South Africa	Case series	14	NS	Median = 35	HIV	Plaques scale crust	Histopathology, Culture, Molecular testing	*Emergomyces africanus*	Amphotericin B, Itraconazole	NS
Rooms et al., [188]	2019	Uganda	Case report	1	F	38	HIV CD4 = 140 cells/Ul	Skin lesions	Histopathology, Gene sequqencing	*E. pasteurianus*	Fluconazole	Favourable
Lochan et al., [189]	2015	South Africa	Case report	1	M	3	HIV	pneumonia, gastroenteritis and herpes gingivostomatitis	Culture and DNA sequencing	*Emmonsia* species	Amphotericin B, Itraconazole	Favourable
Tulleken et al., [190]	2014	South Africa	Case series	3	M	3	HIV CD4 < 5 cells/Ul	skin rash, pneumonia, anemia, and substantial weight loss	Histopathology, Culture	*Emmonsia* species	Amphotericin B, Itraconazole, Fluconazole	Death (n = 2), Favourable (n = 1)

M, Male; F, Female; LTFU, Lost to follow up.

### 3.9. Blastomycosis

Blastomycosis is a fungal infection caused by inhalation of spores of *Blastomyces* species found in soil [191,192]. In Africa, blastomycosis is commonly caused by *Blastomyces percursus* and mostly manifests as pulmonary followed by cutaneous disease, with other organs, such as the brain, being affected [191,192,193,194]. However, extra-pulmonary disease is the commonest manifestation of blastomycosis [195]. Pulmonary disease manifests radiologically with alveolar infiltrates, consolidation, and cavitation as seen on a plain chest radiograph [193]. Bonifaz et al., in their article showed that blastomycosis in middle and east Africa are caused by *B. dermatitidis* with middle and east Africa in the second place of the most endemic areas [191,193]. Blastomycosis is diagnosed using direct microscopy using potassium hydroxide (KOH), culture, histology, antibody test, as well as PCR identification using specimen such as wound secretion, sputum, or bronchial lavage [191]. Blastomycosis is treated using antifungal drugs, such as with amphotericin B for severe disease for at least 12 months and itraconazole, voriconazole, or posaconazole for about 6–12 months for mild to moderate disease [191,192,196,197].

In South Africa, a study involving 20 cases of blastomycosis revealed that most of the cases were caused by *B. percursus* (*n* = 12) and only 8 were caused by *B. emzantsi* [196]. The patients with *B. percursus* had extra-pulmonary disease (*n* = 7), pulmonary disease (*n* = 3), cutaneous disease (*n* = 4), vertebral disease (*n* = 1) and multisystem disease (*n* = 4) [198]. Whereas those with *B. emzantsi* had subcutaneous abscess (*n* = 1) and both pulmonary and vertebral disease (*n* = 2) [198]. The antifungals with the most in vitro potency for the isolates from all 20 cases included voriconazole, posaconazole, itraconazole, amphotericin B, and micafungin [198]. Several cases of cutaneous blastomycosis have been reported in South Africa, Tunisia, and Morocco with no pulmonary involvement [199,200,201,202,203]. *B. dermatitidis* has been implicated in several cases of pulmonary, subcutaneous, vertebral and paravertebral blastomycosis in Tunisia, Morocco, and Tanzania [194,204,205,206]. An earlier case report by Ibrahim et al. described a case of a 37-year-old from Nigeria who was diagnosed with pulmonary blastomycosis with right-sided pleural effusion, was managed with ketoconazole and saline pleural lavage with eventual clinical improvement [207]. In a 70-year retrospective study conducted in Uganda, Kwizera et al. revealed that blastomycosis took up 1.6% of the deep mycoses identified using histology [98], Table 9.

### 3.10. Coccidioidomycosis

Coccidioidomycosis (Valley Fever) is a disease caused by the soil inhabiting spores of the fungi coccidioides. Fisher et al., 2002, isolated *Coccidioides immitis* and *Coccidioides posadasii* as two separate pathogenic species causing Valley Fever. The two are morphologically identical but epidemiologically and genetically varied species [208].

*Coccidioides posadasii* is a soil fungus that is native to certain arid to semi-arid areas of southwestern United States, northern parts of Mexico and South America, while *C. immitis* is endemic to the San Joaquin Valley of California [209,210]. With a geographic overlap between the two species in Southern California [210].

Coccidioidomycosis is transmitted via inhalation of the airborne spores of the dimorphic fungi *Coccidioides* (*C. immitis* and *C. posadasii*) from the soil. The fungus is most frequently acquired in summer or late fall seasons. These dry months are when soil is disturbed by wind and storms. Exposure to contaminated balls of cotton or other fomites can result in infection beyond the endemic region, although rarely [210].

It has an extremely low prevalence outside America, and the few cases reported from Asia and Europe have been seen in travellers to these endemic areas and become clinically significant when they return home or the disease many be transported through contaminated materials [211]. It is also hypothesized that cases of coccidioidomycosis exist in Africa, but had not yet been reported because of failure in diagnosis and misdiagnosis because of its similarity with other pulmonary infections [212]. Person to person transmission of pulmonary infection has not been reported.

Coccidioidomycosis is usually asymptomatic in healthy patients. This occurs in about 60–65% of the patients. The primary infection is in the lungs. The disease can have an acute, chronic, or disseminated form. Acute pulmonary coccidioidomycosis is usually mild, with few or no symptoms, such as fever, sore throat, cough fatigue, etc. It has an incubation period of 7–21 days and is self-limiting in immunocompetent individuals [208].

Chronic pulmonary coccidioidomycosis can develop 20 or more years after initial infection and may present as lung abscess, empyema, bronchopleural fistula, or scarring (fibrosis). While extrapulmonary disseminated infections result of hematogenous spread and are common in, but not limited to, individuals with compromised immune status [213]. Immunosuppressive conditions, such as HIV infection, diabetes mellitus, or malignancy. It may develop weeks, months, or years after the primary infection. It may involve any body organ but has a predilection for dissemination to skin, soft tissue, joints, and the central nervous system. Cutaneous coccidioidomycosis may from puncture with a contaminated object [212,213].

Diagnostic workup for coccidioidomycosis requires detailed history and physical examination, followed by imaging studies with a chest X-ray finding of primary pulmonary disease, including variable nonspecific infiltrates, hilar adenopathy, and pleural effusions. Findings such as cavities and nodules demonstrate progression towards the complicated or residual stage of pulmonary coccidioidomycosis. Confirmation of coccidioidomycosis relies on isolation of fungus in culture, identification upon histopathology or serologic testing [214,215]. Pathological diagnosis requires the demonstration of endosporulating spherules or endospores.

Patients with coccidioidomycosis are usually asymptomatic and only require supportive care. Management of symptomatic patients is according to clinical syndrome [216,217]. The mainstay of antifungal treatment generally consists of amphotericin B deoxycholate or azoles. Amphotericin B is for the severe form of coccidioidomycosis, while azoles, e.g., Ketoconazole, Fluconazole, and Itraconazole, are for mild form of the disease. The duration of therapy is long and may take months to years in certain patients [217].

Only four cases of coccidioidomycosis were identified using histology in Africa. All four cases were caused by unidentified Coccidioides species. None of the cases had a correct clinical diagnosis [98]. Yoo and colleagues [212], described a case of a 23-year-old HIV seronegative Ugandan man with a 10-month history of haemoptysis and difficulty breathing, and a 6-month history of localized swellings on the extremities, associated weight loss, and drenching sweats. He reported no history of travel out of Uganda. Bronchoscopic examination showed two masses occluding the right main bronchus. Bronchoscopic biopsy showed findings consistent with coccidioidomycosis. The patient improved with antifungal treatment and was discharged [212], Table 10. Details of other cases were not found by these authors.

### 3.11. Paracoccidioidomycosis

Paracoccidioidomycosis is a systemic mycosis caused by thermally dimorphic fungi: *Paracoccidioides brasiliensis* and *Paracoccidioides lutzii.* It occurs in the subtropical humid areas of most of the countries in Latin America (Brazil, Argentina, Colombia, and Venezuela) and parts of Central America [218,219,220].

The lungs are the primary site of infection, this is transmitted by inhalation of conidia and mycelial fragments. Infection is usually asymptomatic, however, there are two forms of paracoccidioidomycosis: an acute/subacute form, this is also known as juvenile paracoccydioidomycosis and chronic form or adult paracoccidioidomycosis [221].

The acute or subacute clinical forms, representing 10% of the clinical cases, are prevalent in children and adolescents (younger than 16 years), affecting both sexes equally. Clinical features usually include lymphadenopathy, hepatosplenomegaly, fever, weight loss, malaise, and multiple skin lesions. Mucous membranes and respiratory symptoms are unusual [221].

The chronic form is prevalent in adults (older than 16 years), with a male to female ratio of 20:1, and this difference might be secondary to inhibition of mycelial-to-yeast conversion by oestrogens. They have: primary lung infection, cough, dyspnoea, fever, weight loss, sequelae of chronic pulmonary disease, fibrosis, bullae, and emphysematous changes. Other features include: mucous membrane involvement, oral lesions, cutaneous lesions, and cervical lymphadenopathy. The risk factors for paracoccidiodomycosis include agricultural work, malnutrition, smoking, and alcoholism [221].

Diagnosis of paracoccidioidomycosis is made by microscopy. Rounded, thick-walled yeast cells (typically 15–30 μm in diameter, and up to 60 μm in some cases) with multiple buds (ship wheel-like, pilot wheel-like, or Mickey Mouse ear-like cells) are diagnostic features. Most patients in endemic areas are diagnosed using non-invasive testing, such as serological testing. Immunodiffusion assays (IMMY, Norman, OK, USA) are the most widely used reference assay. This assay is inexpensive and has a high specificity (>95%) and sensitivity (around 80%) [221,222].

Itraconazole has largely been used for patients with (trimethoprim–sulfamethoxazole) has shown itraconazole to be formulation, maintenance treatment with an azole derivative or co-trimoxazole is required [223,224]. Itraconazole (200 mg daily for 9–12 months) is the therapy of choice for patients with mild-to-moderate forms of paracoccidioidomycosis, with co-trimoxazole (for 18–24 months) being the main therapeutic alternative to itraconazole. A short (2–4 weeks) induction therapy with amphotericin B is reserved for severe cases, or for patients who are immunocompromised. Induction therapy with amphotericin B should be followed by 200–400 mg of itraconazole. Surgical management includes relive of granuloma induced spinal cord compression to alleviate fibrotic sequelae [223,224].

A case of a 35-year-old Hausa female from Kano area presented with infiltrated and enormously enlarged and unevenly eroded lips with regional adenopathy. No visceral affection was found. *Paracoccidioides brasiliensis* was cultured and characteristic spherule budding cells were also found in sections taken from the lip and from a cervical lymph node. The patient responded to long acting Sulphormethoxine (Fanasil) [225].

### 3.12. Chromoblastomycosis

Chromoblastomycosis is a subcutaneous mycoses caused by several dematiaceous fungi and was classified together with mycetoma by the WHO as a neglected tropical disease. It is more prevalent in tropical and sub-tropical areas and often associated with poverty. The main etiological agents are *Fonsecaea* spp., *Cladophialophora* spp., and *Phialophora* spp. Chromoblastomycosis is also an implantation mycosis and transmitted via transcutaneous inoculation of spores of the fungi. The causative pathogens are common in the soil and vegetation. Activities that tamper with ecological niche of the fungi, including farming and gardening increases the risk of exposure to infection. Most patients affected by chromoblastomycosis dwell in rural settings. The clinical manifestation is appearance of cutaneous or subcutaneous lesions on the limbs, face, and neck.

Chromoblastomycosis has been reported from every continent, but majority of cases are from South and Central America, Asia, and Africa. In Africa, the present epidemiological data mainly comprise case reports and series. The hotspot on the African continent is Madagascar, where beyond large number of case studies, prospective, and surveillance studies have been undertaken, including recent studies [226,227,228]. In a recent survey evaluating the global burden of chromoblastomycosis from 1947 to 2018, Africa had the second largest number of reported cases after South America, recording 1875 cases from 22 countries [229]. Subsequently, over 80 more cases have been described in Madagascar, Ethiopia and Uganda [98,228,230]. Details on cases of chromoblastomycosis from Africa are summarized in Table 11.

Madagascar had the largest burden of chromoblastomycosis in Africa, followed by South Africa. The disease was noted to be rare in desert areas and the West African sub-region had the least number of cases. The causative pathogen varied depending on geographical location but *F. pedrosoi* predominates, followed by *Cladophialophora* spp. and *Phialophora* spp. However, new studies in Madagascar reported *F. nubica* as the predominate cause of chromoblastomycosis. Like other implantation mycoses, males were more affected than females. Laboratory diagnosis mostly involves traditional direct microscopy, histology, and/or culture. In a recent surveillance study in Madagascar, molecular and MALDI-TOF techniques were employed in confirming diagnosis in suspected cases [227]. The treatment of chromoblastomycosis was not completely documented and absent in most cases. However, treatment was observed to comprise a combination of antifungal therapy, surgical excision, and physical methods. The common antifungal used was itraconazole and occasionally fluconazole, ketoconazole, and potassium iodide. Follow-up information was rarely available. In a current study with substantial treatment and follow-up data in Madagascar, patients were treated with for 4–26 months depending on disease severity and although complete cure or healing was not achieved there was either a major or minor clinical improvement.

### 3.13. Sporotrichosis

Sporotrichosis is a sub-acute to chronic infection caused by the thermal dimorphic fungi of the genera *Sporothrix*. It is more common in tropical and sub-tropical areas. It is usually transmitted through traumatic implantation facilitating the entry of spores into a host and known as an implantation mycosis. The clinical manifestation is broadly classified into skin, mucosal, systemic, and immunoreactive forms [231]. Cutaneous/subcutaneous and lymph node lesions are the commonest manifestations and was observed in disseminated cases. Sporotrichosis is rarely life threatening but may be associated with significant morbidity and reduced quality of life. The ecological niche of the fungus in the environment is mostly in soil and decaying vegetation. Infection is generally initiated during activities such as farming, gardening, animal husbandry, and mining activities [232,233,234]. Zoonotic transmission is common. Sporotrichosis was recently adopted as one of the deep mycoses listed as a neglected tropical disease. The ecology and epidemiology of sporotrichosis vary across different geographical regions or continent. In Africa, the epidemiology of sporotrichosis is not extensively studied except in few countries, such as South Africa and Madagascar [226,234,235]. In South Africa, undocumented cases are estimated to be over 3300, and have been associated with outbreaks among workers in mining settings during the 20th century [236]. In other African countries, there have been sporadic cases reports or series. Details of cases of sporotrichosis in Africa are shown in Table 12. Experts suggest many cases are misdiagnosed and many other cases are undocumented or unpublished. Emerging concerns presently are the effect of climate change on the dynamics of deep fungal infections, such as sporotrichosis [237].

**Table 12 jof-08-01236-t012:** Published cases of sporotrichosis in Africa.

Year	Country	Manifestations	No. of Case(s)	Aetiology	Diagnostic Tool	Treatment	Outcomes	Authors
2015	South Africa	Cutaneous	17	*Sporothrix schenckii*	Culture, Histopathology	-	-	Govender et al. [234]
1927	South Africa	Abscess, Ulcer	14	*Sporothrix beurmanni*	Culture	-	-	Pijper et al. [238]
1963	South Africa		5	*Sporothrix schenckii*	-	-	-	Lurie et al. [236]
1965	Egypt	Superficial, Lymphocutaneous, Disseminated	7	*Sporothrix schenckii*	Culture	Potassium iodide, saline, Lugol’s iodine	Favourable (n = 6), Death (n = 1)	El-mofty et al. [239]
1969	South Africa	Disseminated	1	*Sporothrix schenckii*	Culture	-	-	Brandt et al. [240]
1977	Malawi	Pulmonary	1	*Sporothrix schenckii*	Histopathology, Culture	-	-	Berson et al. [241]
1978	Sudan	Lymphocutaneous	2	*Sporothrix schenckii*	Histopathology	Potassium iodide	Favourable	Gumaa et al. [242]
1978	Zimbabwe	-	3	*-*	-	-	-	Ross et al. [243]
1992	South Africa	Cutaneous, Lymphocutaneous	5	*Sporothrix schenckii*	Culture	Terbinafine	Favourable	Hull et al. [244]
2002	Tanzania	Lymphocutaneous	1	*-*	Histopathology	Potassium iodide	Favourable	Ponnighaus et al. [245]
2008	Morocco	Lymphocutaneous	1	*-*	-	-	-	Benchekroun et al. [246]
2020	Uganda	-	1	*Sporothrix* spp.	Histopathology	-	-	Kwizeraet al [98]
2016	Madagascar	Cutaneous, Lymphocutaneous	34	*Sporothrix schenckii*	Histopathology, Microscopy, Culture, PCR	-	-	Rasamoelina et al. [226]
2016	Zambia	Disseminated, Cutaneous	1	*-*	Histopathology	Itraconazole	Favourable	Patel et al. [247]
2019	Madagascar	Cutaneous	63	*Sporothrix schenckii*	Culture, Molecular testing	-	-	Rasamoelina et al. [235]
1981	Nigeria	Lympho-cutaneous	2	*Sporothrix schenckii*	Culture	-	-	Jacyk et al. [248]
2021	South Africa	Cutaneous, Dissemination	1	*Sporothrix schenckii*	Culture, Histopathology, MALDI-TOF	Fluconazole, Itraconazole	Favourable	Tshisevhe et al. [249]

Sporotrichosis in Africa was largely predominant in South Africa and Madagascar. It affects all age groups but was mostly reported among young adults. Males were the most affected, probably due to occupational and environmental risks of exposure which may be mainly associated with males. The common manifestation recorded was lymphocutaneous lesions that are nodular and frequently ulcerating that mostly affected limbs and faces. Other unusual manifestations do occur, including appearance of muscular, osteoarticular, and visceral lesions [235]. Systemic cases are rarely reported. Sporotrichosis generally occurred in immunocompetent patients, but few cases in immunocompromised patients, such as one in an HIV patient was reported in South Africa [249]. Diagnosis was broadly made by culture and/or histology, and occasionally by direct examination of clinical specimen. The common isolated fungi were *Sporothrix schenckii,* now identified collectively as *S. schenckii sensu stricto* with other closely related variants. In the recent case report from South Africa, MALDI-TOF was employed to confirm culture reports [249]. Potassium iodide was the common treatment option prior to the 20^th^ century. In the past few decades, itraconazole has been the frequently used drug for treatment, but fluconazole or terbinafine is occasionally used when the latter is not available. Outcomes are mostly non-fatal.

## 4. Limitations

Our review focused on commonly encountered IFDs in Africa. Other fungal diseases, including dermatophytosis, pityriasis, tinea nigra, piedra, and mycetoma, were not discussed in this review.

## 5. Conclusions and Future Perspective

In this critical literature review, we exclusively describe the epidemiology of IFDs in Africa, with particular emphasis on the most described IFDs. Cryptococcosis is the most common IFD in Africa, contributing significantly to HIV-related deaths, especially in the high-burden sub-Saharan Africa. Though mostly undiagnosed, invasive aspergillosis is increasingly being reported, mostly in the setting of pulmonary tuberculosis, with a similar predilection towards people living with HIV. Similarly, histoplasmosis previously considered to be non-endemic in Africa, is increasingly being reported, particularly in PLWH. The burden of PCP has significantly reduced owing to increased uptake of anti-retroviral therapy among people living with HIV both in Africa, and globally. Other rare IFDs, such as mucormycosis, talaromycosis, emergomycosis, blastomycosis, and coccidiomycosis have also been described. It is significant to note that there is emerging resistance to most of the available antifungal drugs that are available in Africa.

Our review continues to affirm that IFDs are much more common than expected and contribute to significant mortality and morbidity in Africa. A lot of investments have been made on cryptococcal meningitis, a leading cause of mortality in people living with HIV/AIDS in Africa, yet other invasive fungal diseases, such as invasive aspergillosis and histoplasmosis, are still neglected. We recommend a considerable investment into clinical research, diagnostics, and management of these fungal diseases, that cause significant morbidity and mortality. Important areas of research include diagnosis and therapeutics, tailored to the low- and middle-income nature of most African countries.

## Figures and Tables

**Table 2 jof-08-01236-t002:** Prevalence of cryptococcal disease across Africa.

Country	Pub Year	Study Design	Study Period	Study Pop	Sample Size	CD4 Mean/ Median	ART Status	CrAg Prevalence (N°)	CM Prevalence (N°)	References
Togo	2017	Retrospective and descriptive	2006–2016	Hospitalized HIV infected patients	8025	65 ± 22	83% on ART	NA	1.5 (102/8025)	Wateba et al., [61]
Botswana	2019	Cross sectional	2000–2015	Hospitalized patients with meningitis	21560	91 (37–216)	47% on ART	NA	89(4432/5004)	Tenforde et al. [59]
Ghana	2011	Retrospective	2008–2009	Advanced HIV out- patients	92	28 (8–54)	Naïve	2 (2/92)	NA	Mamoojee et al. [78]
Ghana	2022	Cross sectional	2020–2021	Adult HIV-infected outpatients	150	1049.1 (258.4–1480.6)	52% on ART	2.7% (4/150)	100(3/3)	Ocansey et al. [12]
Ghana	2012	Retrospective	2008–2010	Patients suspected of meningitis	163	NA	NA	NA	11.7 (19)	Owusu et al. [73]
Sierra Leone	2020	Cohort	2018	Adults HIV patients	170	45 (23–63)	44% on ART	4.7 (8/170)	62.5 (5/8)	Lakoh et al. [11]
Senegal	2013	Retrospective and prospective	2004–2011	Hospitalized patients with meningitis	1342	27 (1–375)	35.8% on ART	NA	7.8 (106/1342)	Sow et al. [60]
Senegal	2016	Cross sectional	2009–2013	Hospitalized adults HIV patients	541	102 ± 165	33.5 on ART	9.2(50/541)	34 (17/50)	Manga et al. [19]
Uganda	2013	Cohort	2009–2010	HIV positive patients	563	51 (16–171)	18.4% on ART	5.7(32/563)	NA	Andama et al. [53]
Uganda	2012	Cross sectional	2009–2010	HIV infected adults	367	23 (9–51)	NA	19(69/367)	6.5 (24/367)	Oyella et al. [22]
Mali	2008	Prospective	2001–2002	Hospitalized patients with meningitis	204	NA	NA	NA	8.3 (17/204)	Oumar et al. [76]
Mali	2011	Prospective	2003–2004	Hospitalized patients with meningitis	569	NA	NA	NA	2.5 (14/569)	Minta et al. [13]
Kenya	2010	Prospective and observational	2008–2009	HIV suspected CM patient	340	41 (2–720)	29.7% on ART	NA	33 (111/340)	Mdodo et al. [18]
Mozambique	2020	Retrospective	2018–2019	Hospitalized HIV patients	1795	79 (31–193)	53.7% on ART	7.5 (134/1795)	71.6 (96/134)	Deiss et al. [62]
Tanzania	2011	Cohort	2009–2010	HIV outpatients	333	209 (87–278)	49.3% on ART	5.1 (17/333)	4.4 (15/333)	Wajanga et al. [17]
DRC	2021	Retrospective	2015–2017	Hospitalized HIV patients	4283	168.7 ± 137.1	35.2 on ART	NA	2.8 (108/4283)	Katabwa et al. [63]
DRC	2020	Descriptive	2011–2014	Hospitalized HIV patients with meningitis	261	79 (66–105)	NA	NA	8.8 (23/261)	Zono et al. [72]
DRC	2021	Retrospective and descriptive	2018	Hospitalized HIV patients	1877	NA	NA	NA	21.8 (409/1877)	Ngoy et al. [75]
South Africa	2015	Retrospective	2009–2010	HIV infected patients	1494	NA	NA	2.1 (30/1460)	NA	Govender et al. [64]
Ethiopia	2017	Cross sectional	2016–2017	HIV infected patients	267	NA	52% on ART	3.4 (9/267)	NA	Hailu et al. [59]
Ethiopia	2019	Cross sectional	2017	HIV infected patients	183	434.4 ± 286.3	All on ART	7.7 (14/183)	NA	Geda et al. [58]
Ethiopia	2021	Cross sectional	2019	Hospitalized HIV patients	140	NA	50% on ART	11.43 (16/140)	NA	Jemal et al. [57]
Ethiopia	2020	Cross sectional	2018–2019	HIV infected outpatients	200	54 (2–97)	73.5% on ART	4 (8/200)	NA	Negash et al. [56]
Burkina Faso	2012	Retrospective	2002–2010	Hospitalized patients with meningitis	5129	56 (13–387)	NA	NA	1.8(61/5129)	Bamba et al. [39]
Nigeria	2016	Cross sectional and prospective	2012–2014	HIV infected patients	432	74 (6–1264)	Naïve	1.6 (7/432)	NA	Bologun et al. [9]
Nigeria	2021	Case control	NA	HIV positive and HIV Negative outpatients	342	NA	NA	8.5 (29/342)	NA	Odegbeni et al. [51]
Nigeria	2021	Cross sectional	2017–2018	Hospitalized patients with meningitis	184	32.5 (8–109)	NA	NA	16.8 (31/184)	Okolo et al. [10]
Nigeria	2016	Retrospective, cross- sectional	2004–2016	HIV infected outpatients	2752	NA	Naïve	2.3 (63/2752)	NA	Ezeanolue et al. [47]
Nigeria	2016	Cross sectional	2014–2015	Adult HIV-infected outpatients	214	160 (90–210)	95.3% on ART	8.9 (19/214)	NA	Oladele et al. [14]
Nigeria	2017	Cross sectional	2016	HIV infected patients	215	NA	Naïve	16.7 (37/215)	NA	Goni et al. [24]
Nigeria	2020	Cross sectional	2014–2017	HIV infected patients	300	NA	NA	19.67 (59/300)	25.4(15/59)	Ezenabike et al. [50]
Nigeria	2017	Cross sectional	2016–2017	HIV positive patients	326	NA	81.3% on ART	11 (36/326)	NA	Mohammed et al. [23]
Nigeria	2012	Cross sectional	2011	HIV infected outpatients	150	NA	Naïve	12.7 (19/150)	NA	Osazuwa et al. [49]
Nigeria	2019	Cross sectional	2018	HIV infected patients	290	NA	NA	1.4 (4/290)	NA	Chukwuanukwu et al. [38]
Nigeria	2010	Cross sectional	NA	HIV infected patients	100	89 ± 60	NA	NA	36 (36/100)	Gomorep et al. [52]
Cameroon	2018	Cross sectional	2015–2017	HIV infected outpatients	186	44 (27–75)	Naïve	23.1 (43/186)	21.7 (5/23)	Temfack et al. [55]
Cameroon	2013	Cross sectional	2004–2009	Hospitalized HIV infected patients	672	23 (10–61)	NA	NA	11.2 (75/672)	Luma et al. [65]
Cameroon	2021	Cross sectional	2018	HIV infected children	147	NA	96.60 on ART	6.12 (9/147)	NA	Kalla et al. [67]
Cameroon	2015	Cross sectional	2009–2011	HIV infected patients with Signs of meningitis	146	NA	NA	NA	28.08 (41/146)	Ngouana et al. [66]
Cameroon	2020	Retrospective and descriptive	2010–2018	Hospitalized children with meningitis	331	29 (10–100)	NA	NA	3.6 (12/331)	Nguefack et al. [69]
Cameroon	2012	Cross sectional	2010	HIV positive patients	105	NA	NA	NA	9.86 (29)	Dzoyem et al. [68]

NA: not available/not applicable.

**Table 3 jof-08-01236-t003:** Observational studies and case series on histoplasmosis from Africa.

Country	Authors	Study Design	Number of Males/Females	Study Size/ Population	Age	Affected Site	Number of Cases (%)	Diagnostic Tool	Incidence
Nigeria	Oladele et al., 2022 [82]	Cross-sectional	377 males, 611 females	988	Median age-39-year-old	Lungs, skin	76 (7.7%)	*Histoplasma* urinary antigen assay	-
Nigeria	Ekeng et al., 2022 [80]	Descriptive cross-sectional	114 males, 119 females	213	Mean age-39-year-old	Lungs	27 (12.7)	*Histoplasma* urinary antigen test and/or PCR	-
Nigeria	Lucas 1970 [91]	Case series	NS	-	10 months–65 years	Skin, bones, subcutaneous	-	Histopathology	-
Ghana	Ocansey et al., 2022 [12]	Prospective cross-sectional	41 males, 109 females	150	Age range: 18–62	NS	5 (4.7)	*Histoplasma* urinary antigen test, histopathology	-
Tanzania	Lofgren et al., 2012 [92]	Retrospective	323 males, 647 females	970	Median age: 31	NS	9 (0.9)	*Histoplasma* antigen assay in serum and urine	-
Cameroon	Kuate et al., 2021 [81]	Descriptive cross-sectional	37 males, 101 females	138	Mean age: 43.7	Lungs, skin	36 (26)	*Histoplasma* urinary antigen test	-
Cameroon	Mandengue et al., 2015 [93]	Cross-sectional study	NS	56	NS	Lungs, bronchus, skin	7 (13)	Histopathology	-
Republic of Congo	Amona et al., 2021 [94]	Retrospective (Case series)	30 males, 14 females, unclear in the remainder	-	Mean age-24 years, Median age 22 years	Skin, lymph nodes, bones	57	Histopathology, *Histoplasma* antigen assay, Direct examination	1–3 cases each year
Togo	Darre et al., 2017 [95]	Retrospective and descriptive	11 males, 6 females	-	Mean age-27.2	Skin, mucosa, bones, ganglion	17	Histopathology, Culture, Microscopy	1.1
DRC	Pakasa et al., 2018 [96]	Case series	13 males, 23 females	-	Median age-20.5 years	Skin, lymph nodes, bones	36	Histopathology, Immunohistochemistry, RT-PCR (*n* = 3)	-
South Africa	Kthali et al., 2021 [97]	Retrospective and descriptive	14 males, 10 females	-	Mean age-34.5, Median age-36.5 years	Skin	24	Histopathology, Culture, PCR	-
Uganda	Kwizera et al. [98]	Retrospective	-		-	Skin	64 (9.2)	Histopathology	-

NS: Not stated.

**Table 4 jof-08-01236-t004:** Cases of IA that have been reported from Africa.

Authors	Country	Study Type	Sex/Age/ Number of Cases	Clinical Presentation	Comorbidity	Diagnostic Tool	Aetiology	Clinical Types	Treatment	Outcomes
Northern Africa										
Bakhti et al. 2015 [107]	Algeria	Case report	F/7	Subcutaneous abscess on the chest and right arm, seizure, intracranial hypertension	Chronic granulomatous disease	Serology, Histopathology, CT scan	*A. nidulans*	Invasive disseminated aspergillosis with intracranial localization	VRZ	D
Trabelsi et al. 2013 [108]	Tunisia	Case series	M/48	Fever, repeated pneumopathy	Renal transplant recipients, CMV, DM, broad spectrum antibiotics, renal failure	Serum galactomannan assay, CT scan,	*A. terreus*	IPA	VRZ	Fv
			M/41	Fever, cough, dyspnoea	Renal transplant recipient, CMV, broad spectrum antibiotics, renal failure	Serum galactomannan assay, CT scan, CXR	*A. flavus*		AMB, VRZ	D
El Hakkouni et al. 2018 [109]	Morocco	Case report	M/44	Cough, bloody diarrhoea	HIV	CT scan	*A. fumigatus*	IPA	NS	D
El-Sayed et al. 2012 [110]	Egypt	Prospective	N = 30. F (*n* = 9), M (*n* = 21) 3 mths to 14 years	Fever (*n* = 27), bronchopneumonia (*n* = 3)	SCID (*n* = 2), neutropenia (*n* = 4), broad spectrum antibiotics	PCR	*-*	IPA (*n* = 6)	NS	NS
Gheith 2014 et al. [111]	Tunisia	Prospective	*N* = 175 1–65 years	Cough, chest pain, haemoptysis	Neutropenia secondary to haematological malignancies	Culture (*n* = 23), CT scan (*n* = 11), Serum galactomannan assay (*n* = 23), PCR	*A. niger, A. tubingensis, A. flavus, A. westerdijkiae, A. fumigatus, A. nidulans*	(*n* = 23) IPA, Invasive ethmoiditis with periorbital extension	AMB, VRC or both	D (*n* = 14)
Hadrich et al. 2020 [112]	Tunisia	Prospective	*n* = 105 F (*n* = 13), M (*n* = 16)	NS	AML (*n* = 45), ALL (*n* = 35), medullar aplasia (*n* = 15), other diseases (*n* = 10)	Culture, CT scan (*n* = 18), Serum galactomannan assay (*n* = 29)	*A. flavus, A. fumigatus, A. niger, A. ochraceus, Aspergillus species*	IPA (*n* = 29)	NS	D (*n* = 20), Fv (n = 9)
Eastern Africa										
Ahmed et al. 2018 [113]	Sudan	Case series	F/9	Proptosis, nasal obstruction, headache	NS	Serology, Histopathology, CT scan and MRI	*A. flavus*	Invasive rhinosinusitis with orbital extension	Surgery, ITC, and nasal spray	NS
			F/10		NS	Histopathology. MRI, CT scan	*A. flavus*	Invasive rhinosinusitis with orbital extension	ITC, nasal spray, Surgery	NS
			M/8		NS	Histopathology, CT scan and MRI	*A. flavus*	Invasive rhinosinusitis with orbital extension	ITC, surgery	NS
			M/9		NS	Histopathology, CT scan, MRI, Serology	*A. flavus*	Invasive rhinosinusitis with orbital extension	Surgery, ITC, nasal spray	NS
Kwizera et al. 2020 [98]	Uganda	Retrospective	N = 23	NS	Bronchiectasis (*n* = 2), HIV (*n* = 2), anti-koch’s (*n* = 1)	Histopathology *	NS	IPA (*n* = 8)	NS	NS
West Africa										
Onyekonwu et al. 2005 [114]	Nigeria	Case report	F/60	Nasal blockage, discharge, proptosis, seizure	-	CT scan	*Aspergillus species*	Sino-orbital aspergillosis with CNS complication	Surgery, keto	D
Aleksenko et al. 2006 [115]	Ghana	Case report	M/20	Fever, generalized body pain, ascending stiffness, headache, chest pain, cough, vomiting, diarrhoea, difficulty in breathing	Antibiotics	Histopathology *	*Aspergillus species*	Disseminated IA	NT	D
Southern Africa										
Wong et al. 2012 [116]	South Africa	Prospective	(*n* = 39) F = 19, M = 20. Age-range 32–40 years	Weakness, fever, LAD, pancytopenia	Hodgkin lymphoma, HIV	Histopathology *	ND	IPA (*n* = 1)	NT	D

F, Female; M, Male; Fv, Favourable; D, Death; NS, not specified/not stated; *, post-mortem; IPA, invasive pulmonary aspergillosis; CT, computed tomography; MRI, magnetic resonance imaging; ITC, itraconazole; AMB, amphotericin B; Keto, ketoconazole; VRZ, voriconazole; NT, not treated; ND, not done; mths, months; AML, acute myeloid leukaemia; ALL, acute lymphoblastic leukaemia; HIV, Human immunodeficiency virus; CMV, cytomegalovirus; CR, case report; LAD, lymphadenopathy; SCID, severe combined immunodeficiency; DM, diabetic mellitus.

**Table 5 jof-08-01236-t005:** Published cases of candidaemia in Africa.

Country	Year	Number of Cases	Aetiology	Diagnostic Tool	Treatment	Outcomes	Authors
South Africa	2017	1	*C. auris*	Blood culture	amphotericin B (n = 1)	Death (n = 1)	Lockhart et al.[132]
South Africa	2020	108	*C. albicans* (n = 51*), C. glabrata* (n = 32), *C. parapsilosis* (n = 11*), C. tropicalis* (n = 5), Others (n = 9)	Blood culture	None (n = 31) Fluconazole resistance C. glabrata (n = 35) C. krusei (n = 3)	Death (n = 59)	Hussain et al. [129]
South Africa	2018	48	*C. krusei*	Blood culture	Antifungal therapy (n = 37) Fluconazole and amphotericin B (n = 3) Fluconazole only (n = 1) amphotericin B (n = 2)	Death (n = 7)	Van Schalkwyk et al. [144]
South Africa	2016	2172	*C. albicans* (n = 517), *C.parapsilosis* (n = 488*), C. glabrata* (n = 100), *C. tropicalis* (n = 54) and *C.krusei* (n = 22)	Blood culture	-	-	Naicker et al. [134]
South Africa	2021	2996	*C. parapsilosis* (42%), *C. albicans* (36%)	Blood culture	Fluconazole alone (32%) amphotericin B alone (35%) amphotericin B and Fluconazole (30%)	-	Shuping et al.[138]
Algeria	2020	66	*C. tropicalis* (n = 19) *C. parapsilosis* (n = 18) *C. albicans* (n = 18) *C. glabarata* (n = 6) *C. dubliniensis* (n = 1)	Blood culture	Fluconazole (n = 12) Caspofungin (n = 7) amphotericin B (n = 3) None (n = 21)	Dead (n = 21)	Megri et al [140]
Kenya	2019	224	*C. auris* (n = 77) *C. albicans* (n = 50) Other (n = 74)	Blood culture	-	Death (n = 28)	Adam et al. [136]
South Africa	2013	268	*C. albicans* (n = 123),*C. parapsilosis* (n = 67*), C. glabrata* (n = 58), *C. krusei* (n = 9), *C. tropicalis* (n = 9), *C. guilliermondii* (n = 1) *C. lustitaniae* (n = 1).	Blood culture	-	Death (n = 122)	Kreusch et al. [141]
Egypt	2013	36	*C. albicans* (n = 3) Non-albicans *Candida* (n = 2), others were not specified	Blood culture, Seminested PCR	-	Death (n = 17)	Ramy et al. [143]
South Africa	2019	6669	*C. parapsilosis* (n = 2600) *C. albicans* (n = 1353), *C. auris* (n = 794), *C. glabarata* (n = 598), *C. tropicalis* (n = 140) *C. krusei* (n = 98) Mixed cases (*C.auris* and a non-auris species) (n = 29)	Blood culture	-	-	Van Schalkwyk et al. [145]
Nigeria	2017	27	*C. albicans* (n = 21), *C.krusei* (n = 2) Others (n = 4)	Blood culture	Fluconazole (n = 27)	Death (n = 5)	Ezenwa et al. [142]
South Africa	2022	45	*C. auris* (n = 45)	Blood culture	amphotericin B Echinocandin	Death (n = 19)	Parak et al. [135]
Egypt	2014	88	*C. albicans* (40%) *C. parapsilosis* (25%), *C. tropicalis* (17%), *C. glabarata* (8%)	Blood culture	-	Death (16.7%)	Hegazi et al. [139]
Tunisia	2011	130	*C. tropicalis* (37.7%), *C. albicans* (22.3%), *C. glabrata* (19.2%), *C. parapsilosis* (12.2%).	Blood culture	-	-	Sellami et al. [146]
South Africa	2022	618	*C. albicans* (n = 193), *C. parapsilosis* (n = 82), C*. auris* (n = 72), *C. glabarata* (n = 21), *C. krusei* (n = 54)	Blood culture	-	-	Chibabhai et al. [137]
Tunisia	2012	4	*C. albicans* (n = 3) *C. parapsilosis* (n = 1) *C. krusei* (n = 1) *C. tropicalis* (n = 1)	Blood culture, PCR	Fluconazole and amphotericin B (n = 2) Fluconazole and Voriconazole (n = 1)	Death (n = 2)	Saghrouni et al. [147]

**Table 6 jof-08-01236-t006:** Published cases of Mucormycosis in Africa.

Authors	Country	Number of Males/Females	Number of Cases	Mean Age	Clinical Type	Diagnostic Tool	Outcomes
Elmahallawy et al. 2005 [157]	Egypt	4 males 1 female	5	7.1	Rhinocerebral and pulmonary mucormycosis	Histopathology, Culture	3 Death 1 Survival
Bodenstein et al. 1993 [158]	South Africa	3 males 4 females	7	NS	Rhinocerebral mucormycosis	Histopathology	4 Death 3 Survivals
Hauman et al. [151]	South Africa	2 females	2	7.4	Orofacial mucormycosis	Microscopy	1 Death 1 NS
Zaki et al. 1989 [152]	Egypt	8 males 2 females	10	50.1	Pulmonary Mucormycosis	Histopathology, Culture, PCR	1 Death 9 survivals
Alfishaway et al. 2021 [159]	Egypt	14 males 7 females	21	53.8	Rhinocerebral and pulmonary mucomycosis	Microscopy, Culture, Histopathology	7 Death 14 survivals
Alloush et al. 2022 [160]	Egypt	9 males 5 females	14	64.7	Rhinocerebral Mucormycosis	Microscopy	3 Death 11 survivals
Anane et al. 2009 [161]	Tunisia	8 males 9 females	17	NS	Rhinocerebral mucormycosis	Microscopy	11 Death 6 Survival
Thomson et al. 1991 [162]	South Africa	NS	20	NS	Gastointestinal mucormycosis	Histopathology	7 Death 13 Survival
Kahn et al. 1963 [163]	South Africa	6 males 10 females	16	22	Gastric and abdominal mucormycosis	Histopathology	NS
Feki et al. 2018 [164[	Tunisia	3 males	3	66	Pulmonary mucormycosis	Histopathology, Culture	3 survivals
Madeney et al. 2017 [165]	Egypt	NS	45	8	Rhinocerebral and gastrointestinal mucormycosis	Histopathology, Culture	15 Death 30 survivals

NS: Not stated.

**Table 9 jof-08-01236-t009:** Studies published on Blastomycosis in Africa.

Country	Clinical Type	Number of Cases	Affected site	Aetiology	Diagnostic Tool	Treatment	Outcomes	Year	Authors
South Africa	Cutaneous (*n* = 4), Subcutaneous (*n* = 1), Pulmonary (*n* = 3), Vertebral (*n* = 1), Multisystem (*n* = 6)	20 (1967–2014)	-	*B. persursus, B. emzantsi*	Histopathology, Culture, Microscopy, PCR	Voriconazole, posaconazole, itraconazole, amphotericin B and micafungin	Death (n = 5), Favourable (n = 5), Unknown (*n* = 10)	2020	Maphanga et al. [198]
Tunisia	Pulmonary, Subcutaneous disease	1	Right and left Lungs, Left paravertebral swelling around T10	*B. dermatitidis*	Culture, Radiology	Itraconazole	Favourable	2020	Abdallah et al. [194]
South Africa	Cutaneous disease	1	Scalp, Face, Neck	*B. dermatitidis*	Histopathology, Culture	Itraconazole	Favourable	2012	Motswaledi et al. [200]
Morocco	Pulmonary, Vertebral disease	1	Left lung, Vertebra	*B. dermatitidis*	Radiology, Histopathology	Ketoconazole	Favourable	2012	Rais et al. [204]
Tunisia	Cutaneous disease	2	Right leg (patient 1), Left knee and Left shoulder (patient 2)	*B. dermatitidis*	Histopathology, Culture	-	-	2004	El Euch et al. [199]
Tunisia	Cutaneous disease	1	Right leg	*B. dematitis*	Histopathology, Enzyme linked immunosorbent assay	Itraconazole	Favourable	2017	Ben Salem et al. [201]
Tunisia	Cutaneous disease	3	Right cheek (patient 1), Left cheek (Patient 2), Right iliac fossa	*-*	Histopathology, Culture	Ketoconazole	Favourable	2006	Ferchichi et al. [202]
Tunisia	Pulmonary, Vertebral disease	1	Right lower limb, Right lung	*B. dermatitidis*	Histopathology, Culture, Radiology	Itraconazole	Favourable	2008	Cheikh Rouhou et al. [205]
Tanzania	Pulmonary, Cutaneous disease	1	Lung, Nose, Right hand and right forearm, Left buttock, Left upper arm	*B. dermatitidis*	Histopathology	Itraconazole	Favourable	2006	Alvarez et al. [206]
Nigeria	Pulmonary	1	Right Lung	*B. dermatitidis*	-	Ketoconazole	Favourable	2001	Ibrahim et al. [207]
Morocco	Cutaneous disease	1	Forearm	*B. dermatitidis*	Histopathology	-	Death	2007	Harket et al. [203]
Uganda	Pulmonary, Cutaneous disease	11	Lower limbs (*n* = 7), Upper limbs (*n* = 1), Abdominal wall (*n* = 1)	*B. dermatitidis*	Histopathology	-	-	2020	Kwizera at al [98]

**Table 10 jof-08-01236-t010:** Case studies on Coccidioidomycosis published in Africa.

Authors	Year	Country	Study Type	Number of Cases	Sex	Age	Symptoms	Diagnosis	Treatment
Yoo SD et al., 2020 [212]	2020	Uganda	Case report	1	Male	23	haemoptysis and difficulty breathing, weight loss and drenching sweats localized swellings on the extremities	Histopathology	Intravenous amphotericin B deoxycholate 0.7 mg/kg was given for 10 days daily and followed by oral itraconazole 200 mg/day
Kwizera et al. [98]	2021	Uganda	Retrospective	4	NS	NS	NS	Histopathology	NS

NS: Not stated.

**Table 11 jof-08-01236-t011:** Published cases of chromoblastomycosis in Africa.

Year	Country	Clinical Manifestations	No. of Cases	Aetiology	Diagnostic Tool	Treatment	Outcomes	Authors
2021	Africa	Cutaneous lesions	1875	*Fonsecaea* spp., *Cladophialophora* spp., *Phialophora* spp.	Microscopy, Histopathology, Culture, PCR	Surgical excision, Itraconazole, Fluconazole, Terbinafine, Ketoconazole,	-	Santos et al. [229]
2020	Madagascar	Cutaneous lesions	50	*Fonsecaea* spp., *Cladophialophora* spp.	Microscopy, Histopathology, PCR, MALDI-TOF	Itraconazole	Favourable	Rasamoelina et al. [227]
2021	Ethiopia	Cutaneous lesions	12	-	Microscopy, Histopathology	-	-	Abate et al. [226]
2020	Uganda	-	34	-	Histopathology	-	-	Kwizera et al. [98]

## Data Availability

All underlying data have been included in the main text of the manuscript.

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
