# Peer review of "Invasive Fungal Diseases in Africa: A Critical Literature Review"

_jof, 2022, doi:10.3390/jof8121236_

Round 1
Reviewer 1 Report
Line# 92 “in all other fungal diseases” please mention the name
Why you not mention the date of you searching?
Which guidelines do you follow for searching the articles?
How many researchers extract the data? What about personal bias? How do you minimize it?
It will be better to mention the inclusion and exclusion criteria.
Is there any case report? Did you select that or not?
How you analyze the extracted data? Any software? Any statistical analysis?
The method section needs to be revised, write the complete detail of your searching, and extraction of data till the analysis
How many studies and number of cases you find for each IFD and from which region of Africa? Also mention the risk factors from you literatures.
Please specify your study to Africa region, there is more detail about global epidemiology
Candidemia is mostly nosocomial? Do you find such cases in your studies, please mention that. Also write the preventive measure
Any information about the clade type of C. auris? If available please mention
Did you find any study related to co-morbidity with COVID-19, as you mentioned Mucormycosis, what about the other IFD?
Please mention the limitation and implications of your study.
Author Response
S/N |
REVIEWER COMMENTS |
AUTHORS’ REPLY |
|
REVIEWER 1 |
|
1. |
Line# 92 “in all other fungal diseases” please mention the name
|
Thank you for spotting that error. It has been expunged |
2. |
Why you not mention the date of you searching?
|
No date limitation or any other search criteria were applied, to avoid exclusion of articles on IFDs in Africa. Kindly check lines 88 and 89 |
3. |
Which guidelines do you follow for searching the articles?
|
This was not designed as a formal systemic review |
4. |
How many researchers extract the data? What about personal bias? How do you minimize it?
|
All researchers (coauthors) were involved in data curation, thereafter 3 researchers (FB, BEE, WK) screened publications for eligibility. Lines 96 to 99 |
5. |
It will be better to mention the inclusion and exclusion criteria.
|
Inclusion criteria has been highlighted in lines 92 and 93, exclusion criteria has been highlighted in lines 101 to 102 |
6. |
Is there any case report? Did you select that or not? |
Case reports were particularly highlighted for rare fungal diseases |
7. |
How you analyze the extracted data? Any software? Any statistical analysis? |
There was no statistical analysis. We did a narrative describing the burden of IFIs in Africa. Values of incidence or prevalence are as cited in the referenced studies |
8. |
The method section needs to be revised, write the complete detail of your searching, and extraction of data till the analysis |
The search terms have been highlighted. Kindly check lines 91 – 96. |
9. |
How many studies and number of cases you find for each IFD and from which region of Africa? Also mention the risk factors from you literatures. |
This idea is welcomed but may be conflicting as some case series may have cases previously mentioned in a larger series |
10. |
Please specify your study to Africa region, there is more detail about global epidemiology |
We have done so |
11. |
Candidemia is mostly nosocomial? Do you find such cases in your studies, please mention that. Also write the preventive measure
|
Yes…. clades I to IV. Candidaemia section line 79 |
12. |
Any information about the clade type of C. auris? If available please mention |
Yes…. clades I to IV. Candidaemia section line 74 |
13. |
Did you find any study related to co-morbidity with COVID-19, as you mentioned Mucormycosis, what about the other IFD? |
None |
14. |
Please mention the limitation and implications of your study. |
The limitations have now been highlighted in a paragraph prior to the conclusion. The implications are already stated in the section on conclusion and future perspectives |
Reviewer 2 Report
It is important to raise awareness about fungal diseases in Africa and to review all available information. In general the manuscript is well written but the candidemia section should be rewritten.
Comments
1. Introduction lines 73-75. These lines about the immune response to fungal pathogens are too simplistic. I propose to remove these sentences.
2. Methods line 92. The search strategy is not entirely clear. What do the authors mean with .. ‘and all other fungal diseases AND Africa’? It is also not clear which studies were selected and which ones not (observational studies and case series is not precise enough). Randomised trials as e.g. with respect to cryptococcosis could not be included?
3. Invasive aspergillosis, line 232 prolonged use of broad-spectrum of antibiotics is not an independent risk factor for invasive aspergillosis. Add influenza to the risk factors for this disease. Replace galactomannan by aspergillus antigen as diagnostic tests based on other antigens are now available.
4. Use the abbreviation PLWH (people living with HIV) consistently throughout the text.
5. Pneumocystis pneumonia. Pneumocystis cannot be cultured. The statement that culture is less sensitive gives the impression that culture is a diagnostic tool being used which is not the case.
6. Candidemia.
a. Add C. auris to the list with the commonest Candida species with the info to the countries where this is the case.
b. Beta-D-glucan (BDG) instead of bera-D-glucan
c. Replace the sentences about the sensitivity of the BDG for specific species by some general information about sensitivity for candidemia and mention also that this test is not specific for candidemia.
d. Give general information about resistance to azoles for Candida species instead of mentioning the number of isolates that were resistant to fluconazole in a specific study (3 isolates of C. krusei that were resistant to fluconazole is a meaningless statement as all C. krusei are intrinsically resistant to this drug).
e. ‘Other surveillance based publications showed about 2172 incidence cases of candidaemia’, give some information about this study (region, period,…).
f. The authors mention that mortality of C. albicans is higher than C. auris but this is contradictory to the percentages they are giving (29% and 36% respectively).
g. Mentioning the number of C. albicans isolates (n=123) and (n=21) is not informative without any additional information. Replace by %.
7. Mucormycosis. The authors mention that these fungi closely resembles aspergillosis. What do they mean? The clinical presentation of patients with invasive aspergillosis or mucormycosis is very similar? Discrimination based on histopathology is difficult? On cultures it is quite easy to differentiate both fungi.
8. Page 26. ‘in Africa’ should not be added to the title about sporotrichosis (similarity to the other titles)
9. Conclusion. The statement that emerging resistance to most of the available antifungal drugs is not covered in the text, either this should be removed from the conclusion or information about resistance development should be added to the text.
10. Typo’s, grammatical issues:
a. Line 43. Africa instead of African
b. Line 101. Caused by the fungal species instead of fungi species
c. Page 13 (line numbers is lacking in the text). Pneumocystis jirovecii is a fungus instead of is fungi
d. Rephrase the first sentence of the conclusion ‘the most described diseases’
Author Response
|
Reviewer 2 |
|
1 |
Introduction lines 73-75. These lines about the immune response to fungal pathogens are too simplistic. I propose to remove these sentences. |
We have done so |
2 |
Methods line 92. The search strategy is not entirely clear. What do the authors mean with .. ‘and all other fungal diseases AND Africa’? It is also not clear which studies were selected and which ones not (observational studies and case series is not precise enough). Randomised trials as e.g. with respect to cryptococcosis could not be included? |
The statement …´Ì”and all other fungal diseases and Africa…’ has been removed. The methods section has been written in details |
3 |
Invasive aspergillosis, line 232 prolonged use of broad-spectrum of antibiotics is not an independent risk factor for invasive aspergillosis. Add influenza to the risk factors for this disease. Replace galactomannan by aspergillus antigen as diagnostic tests based on other antigens are now available. |
Thank you for your suggestions. We have done so. Lines 240, 245 and lines 268-269 in the section on aspergillosis |
4 |
Use the abbreviation PLWH (people living with HIV) consistently throughout the text. |
We have done so |
5 |
Pneumocystis pneumonia. Pneumocystis cannot be cultured. The statement that culture is less sensitive gives the impression that culture is a diagnostic tool being used which is not the case. |
The statement has been modified as advised. Lines 2 and 25, section on Pneumocystis pneumonia |
6 |
CaCandidaemia. a. Add C. auris to the list with the commonest Candida species with the info to the countries where this is the case. b. Beta-D-glucan (BDG) instead of bera-D-glucan c. Replace the sentences about the sensitivity of the BDG for specific species by some general information about sensitivity for candidemia and mention also that this test is not specific for candidemia. d. Give general information about resistance to azoles for Candida species instead of mentioning the number of isolates that were resistant to fluconazole in a specific study (3 isolates of C. krusei that were resistant to fluconazole is a meaningless statement as all C. krusei are intrinsically resistant to this drug). e. ‘Other surveillance based publications showed about 2172 incidence cases of candidaemia’, give some information about this study (region, period,…). f. The authors mention that mortality of C. albicans is higher than C. auris but this is contradictory to the percentages they are giving (29% and 36% respectively). g. Mentioning the number of C. albicansisolates (n=123) and (n=21) is not informative without any additional information. Replace by %. |
We have done so. Kindly refer to section on Candidaemia a. Line 65 b. Line 67 c. Lines 67-70 d. e. Lines 107-118 f. Lines 86-87 |
7 |
MMucormycosis. The authors mention that these fungi closely resembles aspergillosis. What do they mean? The clinical presentation of patients with invasive aspergillosis or mucormycosis is very similar? Discrimination based on histopathology is difficult? On cultures it is quite easy to differentiate both fungi. |
This statement has been revised |
8 |
PaPage 26. ‘in Africa’ should not be added to the title about sporotrichosis (similarity to the other titles) |
This has been corrected |
9 |
CoConclusion. The statement that emerging resistance to most of the available antifungal drugs is not covered in the text, either this should be removed from the conclusion or information about resistance development should be added to the text. |
Kindly check section on candidaemia lines 78 - 85 |
10 |
TyTypo’s, grammatical issues: a. Line 43. Africa instead of African b. Line 101. Caused by the fungal species instead of fungi species c. Page 13 (line numbers is lacking in the text). Pneumocystis jirovecii is a fungus instead of is fungi d. Rephrase the first sentence of the conclusion ‘the most described diseases’
|
These has all been corrected |
Reviewer 3 Report
Please, review the legends of Tables, particularly Table 4: What means F in the Outcome; you also used F for Female on the same table.
In page 13 (Pneumocystis pneumonia topic), Maybe it is not completely correct to say that "... culture is less sensitive", since the pathogen is nonculturable.
In page 14, please, correct 1,3-beta-D-glucan ("bera" in not correct).
Also in this same page, C. kurei is incorrect as Krusei.
In Table 8, B. dermatitidis is incorrectly written in some places.
Author Response
|
REReviewer 3 |
|
1. |
PL Review table legends especially table 4 |
We have done so |
2 |
Page 13 |
The statement on P jirovecii cultures has been deleted |
3 |
Page 14 |
1,3-beta-D-glucan corrected and C. krusei correctly spelt |
4 |
Table 9 |
B. dermatitidis corrected |